# *Pseudomonas aeruginosa* Cytotoxins: Mechanisms of Cytotoxicity and Impact on Inflammatory Responses

**DOI:** 10.3390/cells12010195

**Published:** 2023-01-03

**Authors:** Stephen J. Wood, Josef W. Goldufsky, Michelle Y. Seu, Amir H. Dorafshar, Sasha H. Shafikhani

**Affiliations:** 1Department of Medicine, Division of Hematology, Oncology and Cell Therapy, Rush University Medical Center, Chicago, IL 60612, USA; 2Department of Surgery, Division of Plastic and Reconstructive Surgery, Rush University Medical Center, Chicago, IL 60612, USA; 3Department of Microbial Pathogens and Immunity, Rush University Medical Center, Chicago, IL 60612, USA; 4Cancer Center, Rush University Medical Center, Chicago, IL 60612, USA

**Keywords:** *Pseudomonas aeruginosa*, infection, virulence factors, cytotoxins, apoptosis, pyroptosis, necroptosis, necrosis, innate immunity, inflammatory responses

## Abstract

*Pseudomonas aeruginosa* is one of the most virulent opportunistic Gram-negative bacterial pathogens in humans. It causes many acute and chronic infections with morbidity and mortality rates as high as 40%. *P. aeruginosa* owes its pathogenic versatility to a large arsenal of cell-associated and secreted virulence factors which enable this pathogen to colonize various niches within hosts and protect it from host innate immune defenses. Induction of cytotoxicity in target host cells is a major virulence strategy for *P. aeruginosa* during the course of infection. *P. aeruginosa* has invested heavily in this strategy, as manifested by a plethora of cytotoxins that can induce various forms of cell death in target host cells. In this review, we provide an in-depth review of *P. aeruginosa* cytotoxins based on their mechanisms of cytotoxicity and the possible consequences of their cytotoxicity on host immune responses.

## 1. Introduction

*Pseudomonas aeruginosa* is one of the most versatile and virulent opportunistic bacterial pathogens described to date [1]. It is a leading cause of bacteremia and sepsis in patients receiving cancer drugs, the most common cause of nosocomial pneumonia, a frequent cause of infections in diabetic ulcers, burn wounds, surgical wounds, and corneal ulcers, and a deadly cause of chronic infection in cystic fibrosis patients [1,2,3,4,5,6,7,8,9,10,11]. Despite aggressive antibiotic therapy, the fatality rate amongst individuals with *P. aeruginosa* infections can reach as high as 40% [1,12,13,14]. These figures have not improved in decades due to the high degree of intrinsic and acquired resistance of *P. aeruginosa* to many antibiotics, as well as the emergence of multi-drug resistant strains [15,16,17,18].

*P. aeruginosa* derives its versatility to cause infections in a broad host range from the large array of virulence factors it possesses. We define virulence factors as factors that are not required for growth in culture media per se. Rather, in their absence, *P. aeruginosa* becomes impaired in its ability to colonize and cause infection in a manner that benefits its survival and/or persistence in vivo. These factors aid the bacterium in colonization and dissemination within the host, protect it against host immune defenses, and/or exacerbate epithelial injury to prevent wound healing, thus maintaining *P. aeruginosa*’s preferred niche, the wound itself [19,20,21].

Induction of cytotoxicity in target host cells is a major virulence strategy that *P. aeruginosa* employs during the course of infection. *P. aeruginosa* has invested heavily in this strategy, as manifested by a plethora of cytotoxins that can induce various forms of programmed cell death in target host cells. Before we categorize these virulence cytotoxins based on their mechanisms of cytotoxicity and the consequences of their cytotoxicity on host immune responses, we will briefly discuss major types of programmed cell deaths.

### 1.1. Apoptosis

In 1965, Lockshin and Williams et al. discovered that during the metamorphosis of the silkworm, specific cells died [22]. They designated this type of death as “programmed” because the same cells died each time. In 1972, Kerr et al. observed a specific type of cell death in human tissues in which the cells exhibited specific morphological characteristics including chromatin condensation (karyorrhexis), nuclear condensation (pyknosis), and fragmentation, cellular shrinkage and fragmentation [23]. They coined this cell death process “Apoptosis,” which means “falling off” like the leaves falling from a tree in autumn. Today, our knowledge of apoptosis has vastly improved. Apoptosis plays fundamental roles in development [24], immune system maturation [25], genome maintenance [26], wound healing [27], protection against autoimmunity and cancer [28,29], and innate immune defenses against pathogens [30]. It has been estimated that out of approximately 37.2 trillion cells in an adult human, 50–70 billion cells (~0.2%) die each day by apoptosis [31], highlighting the importance of apoptosis in human physiology and health. As important as apoptosis is for health, its dysregulation can have dire consequences, leading to many pathological conditions including but not limited to neurodegenerative diseases, autoimmune disorders, impaired infection control, and cancer (reviewed in [32,33]).

At the molecular level, apoptosis is mediated by cysteine aspartate proteases (caspases) which are subdivided into initiator Caspases 2, 8, 9 (mouse and human) and Caspase-10 (human), as well as effector (a.k.a., executioner) Caspases 3, 6, and 7 (mouse and human) [34]. Multiple apoptotic programmed cell deaths (PCD) have been described, but intrinsic (a.k.a., mitochondrial) apoptosis and extrinsic (death receptor-mediated) apoptosis are the most well-characterized apoptotic PCDs [35,36,37] and summarized in Figure 1).

Internal stimuli such as DNA damage trigger intrinsic apoptosis by activating the pro-apoptotic BH3-only subgroup of pro-apoptotic Bcl-2 family proteins (Bim, Bid, Bad, and Puma), which in turn disrupt interactions between the pro-apoptotic Bcl-2 family proteins (Bax and/or Bak) and anti-apoptotic Bcl-2 family proteins (Bcl-2 and BCL-_xL_, BCL-_W_, MCL1, and A1/BFL-1), thus freeing Bax and Bak to mobilize to the mitochondrial outer membrane where their oligomerization is presumed to disrupt the mitochondrial outer membrane, finally resulting in the release of cytochrome *c* [38]. Cytosolic cytochrome *c* then combines with APAF1 and pro-Caspase-9 to form the “Apoptosome,” which activates the initiator pro-Caspase-9 into mature caspase-9 via autoproteolysis, in turn activating effector Caspases 3, 6, and 7 (Caspase-3 is the primary effector caspase) that carry out cell death. Endoplasmic reticulum (ER) and genotoxic stresses can also lead to the activation of the initiator Caspase-2 through the PIDDosome multicomplex (containing PIDD, CRADD/RAID, and pro-Caspase-2) which in turn activates Bid into tBid (truncated Bid) by cleavage, initiates Bax-mediated mitochondrial outer membrane disruption, and eventually leads to Caspase-9 dependent intrinsic apoptosis [39,40,41].

Extrinsic apoptosis involves the activation of so-called death receptors (TNFR1, FasR, DR3, DR4, and DR5) by their cognate external ligands (TNF-α, FasL, TRAIL (Apo2L), and Apo3L) [37,42]. Depending on the specific receptor and ligand, death receptor/ligand interaction results in the recruitment of TRADD, FADD, or DAXX adaptor proteins, leading to the formation of a polymeric activation complex known as DISC (death-inducing signaling complex), which in turn activates initiator Caspases 8 (mouse and human) and/or 10 (human only), subsequently activating the effector Caspases 3, 6, and 7, that carry out the task of cellular demise. Activated Caspases 8 and 10 can further amplify cell death by cross-feeding into intrinsic apoptosis via cleavage of Bid into tBid [43,44].

Apoptotic PCDs are generally believed to be anti-inflammatory in nature for the following reasons. First, the degradation of cytosolic proteins by activated caspases during apoptosis reduces the so-called danger-associated molecular pattern molecules (DAMPs) which are highly pro-inflammatory in nature [45]. Second, DAMPs are not directly released into the environment during apoptosis, thus, they are not available to activate pattern recognition receptors (PRRs) and trigger inflammatory responses in neighboring cells. Rather, they are packaged in membrane-bound apoptotic bodies which are then rapidly cleared by macrophages via phagocytosis [46,47]. Third, uptake of apoptotic bodies suppresses inflammation in macrophages by inhibiting the production of inflammatory cytokines by anti-inflammatory factors such as TGF-β and prostaglandin E2, thus transitioning macrophages into so-called M2 anti-inflammatory phenotype [48,49]. The anti-inflammatory nature of apoptosis is perhaps an important reason why the majority of cytotoxins produced by pathogens employ this mechanism to induce cell death in their target host cells. Apoptosis-inducing cytotoxins in *P. aeruginosa* will be discussed later.

### 1.2. Pyroptosis

Pyroptosis is a form of inflammatory PCD which involves either Caspase-1-dependent canonical inflammasomes or non-canonical inflammasomes driven by Caspase-11 (in mice) or Caspase-4 and Caspase-5 (in humans) [50,51] and summarized in Figure 2. Canonical inflammasomes are multiprotein oligomeric structures formed by ASC adaptor proteins bridging the interaction between NLRP1b, NLRP3, NLRC4, AIM2, or Pyrin (canonical inflammasome subtypes) with pro-Caspase-1 in response to external or internal stimuli (e.g., extracellular adenosine triphosphate (ATP), microbial products, or DAMPs), ultimately culminating in the activation of Caspase-1 though autocleavage and processing [52,53]. In contrast to Caspase-1, which requires the assembly of the aforementioned multiprotein inflammasome complexes for its activation, Caspases 11, 4, and 5 do not require multiprotein complexes for self-processing and activation. Rather, it appears that their activation results from the direct interaction between their CARD domains with intracellular LPS through its lipid A moiety [54,55,56]. Activated Caspases 1, 11, 4, and 5 then drive the cleavage of the pro-pyroptotic factor Gasdermin D (GSDMD), which then oligomerizes to form pores in the plasma membrane causing cell death [55,56,57]. It is worth noting that in contrast to pro-Caspase-1 which is expressed in resting cells, pro-Caspase-11 is not expressed in resting cells and its expression requires priming by LPS which triggers a signaling pathway involving Toll-like receptor 4 (TLR4)/MyD88/NF-κB [58].

Pyroptotic cell death is highly pro-inflammatory in nature not only because cellular demise by this mechanism is associated with the release of cellular contents and cytosolic DAMPs through GSDMD-generated pores in the plasma membrane [55,56], but also because of the processing and release of pro-inflammatory cytokines IL-1β and IL-18 by active Caspase-1 [52,53]. Noncanonical Caspases 11, 4, and 5 do not directly process pro-IL-1 and pro-IL-18 pro-inflammatory cytokines into their active forms (IL-1β and IL-18) by themselves, but they do so indirectly via NLRP3 canonical inflammasome subtype activation by promoting K^+^ efflux in a manner that is dependent on the TRIF adaptor protein [59,60,61,62,63,64].

Because of its inflammatory nature, pyroptosis may be viewed as an altruistic form of cellular sacrifice that is intended to limit infection and spare uninfected neighboring cells through alert signals and inflammatory mediators (IL-1β, IL-18, and DAMPs). In line with this notion, CIAS1 7 unit repeat genetic polymorphism in NLRP3 has been associated with decreased IL-1β levels and increased occurrence of vaginal Candida and Mycoplasma infections [65,66]. Conversely, in cases where the invading pathogen infects and resides within host immune leukocytes, pyroptosis may be beneficial to the pathogen. For instance, polymorphisms in CARD8 (C10X) and NLRP3 (Q705K), both of which are associated with increased NLRP3 inflammasome activity, have shown significant association with increased extrapulmonary *Mycobacterium tuberculosis* infection [67].

It is worth noting that exuberant inflammasome activity has been associated with many pathological conditions including several immunological disorders (such as familial cold auto-inflammatory syndrome (FCAS), cryopyrin-associated periodic syndromes (CAPS), Crohn’s disease, and ulcerative colitis [68,69,70,71,72]); neurological disorders (such as, progressive multiple sclerosis, age-related macular degeneration, Alzheimer’s disease) [73,74,75]; cardiovascular diseases (such as, atherosclerosis, atrial fibrillation) [76,77,78], cancer [79,80,81]; and insulin resistance and type 2 diabetes [82,83,84].

### 1.3. Necroptosis

For decades, necrosis was mostly viewed as an “accidental cell death” that occurred as a result of extreme physicochemical insults such as heat or membrane damage, but this view began to change with the seminal reports published by Holler et al. [85] and Degterev et al. [86], in which they described an alternative Caspase-8-independent non-apoptotic pathway of cell death triggered by Fas ligand (FasL), mediated by Receptor-Interacting Protein Kinase (RIPK), and inhibited by necrostatin-1. Today, necrotic cell death is divided into 2 subcategories, namely “necrosis (a.k.a., oncosis)” which is unregulated, instantaneous accidental cell death as the direct result of massive cellular and/or membrane damage caused by extreme physicochemical conditions, and “necroptosis” a form of regulated, protein-based cell death occurring in response to signaling cues and milder physicochemical conditions [87,88].

Morphologically, necrosis (oncosis) and necroptosis manifest similar features including loss of plasma membrane integrity, cellular and organelle swelling, and the release of cytoplasmic contents into the surrounding environment [89,90]. Mechanistically, necroptosis is initiated by TNF family receptors [85,86,91,92]; by pattern recognition receptors (PRRs) including Toll-like receptors (e.g., TLR3 and TLR4), and by cytosolic viral sensory pathways including retinoic acid-inducible gene-I-like (RIG-I), stimulator of interferon gene (STING), and DNA-dependent activator of IFN-regulatory factors (DAI) [93,94,95,96,97,98] and summarized in Figure 3. Necroptosis is ultimately executed by mixed lineage kinase domain-like (MLKL) protein, which upon phosphorylation at multiple sites by RIPK3 [99,100,101], is mobilized to the plasma membrane through interaction between its exposed N-terminal 4-helix Bundle (NB) domain with phosphatidylinositol phosphates (PIP) within the inner leaflet of the plasma membrane, where it carries out cellular demise by homotypic oligomerization and pore-formation within the plasma membrane in a manner that is assisted by the cytosolic chaperone Hsp90 (heat shock protein 90kDa) [101,102,103,104,105,106]. RIPK3 itself is recruited to a pro-cell death multiprotein cytosolic complex known as “necrosome” via its RIPK homology interaction motif (RHIM) domain interacting with either RIPK1 (in case of TNF family receptors [107]), TRIF (in case of TLRs [108,109]), or directly with DAI cytosolic viral sensor itself through its RHIM domain [96].

It is worth noting that upon recruitment to TNF family receptors (TNFRs), RIPK1 can exist in either “closed” or “open” conformations, which can drastically affect cellular survival and responses to TNF-α. In the “closed” conformation, RIPK1 is ubiquitinated by cIAP1/2 at TNFR which keeps RIPK1 locked in association with TNFR and suppresses its kinase activity [110], allowing for the recruitment of NEMO, IKKα, and IKKβ, collectively known as IkB kinase complex [111]. These interactions then culminate in NF-kB and MAPK activation which in turn drive the cell toward “pro-inflammatory” and “survival” phenotypes, due to NF-kB and MAPK activities, respectively [112,113,114]. In its kinase active “open” conformation, RIPK1 is not ubiquitinated and dissociates from TNFR and recruits and activates RIPK3 in necrosome, which in turn drives the cell toward pro-inflammatory necroptosis cell death, mediated by MLKL [115,116]. It should further be noted that necroptosis cannot occur unless extrinsic apoptosis is inhibited because Caspase-8 in association with cFLIP and FADD (occurring during the extrinsic apoptosis initiation by death receptors engagement with their ligands as discussed above) can recruit both RIPK1 and RIPK3 to a complex dubbed as “Ripoptosome” where they are degraded by Caspase8 [117,118]. Caspase-8 also targets the deubiquitinase cylindromatosis CYLD which further prevents RIPK1 initiation of necroptosis [119].

Similar to pyroptosis, necroptotic programmed cell death is also highly pro-inflammatory in nature because MLKL-mediated pore formation in the plasma membrane leads to the release of cytoplasmic DAMPs [120]. Moreover, MLKL, RIP1, and/or RIP3 have also been shown to activate Caspase-1-dependent canonical inflammasomes, (particularly NLRP3 inflammasome subtype), leading to the processing and release of pro-inflammatory cytokines IL-1β and IL-18 [121,122,123]. Because of its pro-inflammatory nature, necroptosis for the most part benefits the host and is detrimental to invading pathogens. For example, necroptosis has been shown to play an important role against many invading pathogens including but not limited to, *Staphylococcus aureus*, *Yersinia pestis*, human herpes simplex virus (HSV), and vaccinia virus [124,125,126].

Ironically, necroptosis can also benefit pathogens in some cases and be detrimental to the host. For example, necroptosis has been shown to enhance infection in the case of the *Salmonella Typhimurium* [127] and *S. aureus* in the airway lung infection models [128]. As is the case for pyroptosis, exuberant necroptosis can also have dire pathological consequences. Necroptosis has been shown to cause lethal lung damage, while its inhibition has been demonstrated to protect against lung injury and improve survival in a neonatal sepsis mouse model [129]. Necroptosis has also been shown to drive *Listeria monocytogenes* septicemia-associated acute hepatic injury, while its inhibition by RIPK1 deletion protects against this pathology [130]. Necroptosis has been reported to drive systemic inflammatory response syndrome (SIRS) [131], while necroptosis inhibition by genetic means (e.g., deletion in RIPK3 gene) or by pharmacological means (RIPK1 inhibition by necrostatin-1) protects against SIRS [131]. In addition, necroptosis has been implicated in various inflammatory disorders such as Crohn’s and ulcerative colitis [132,133], rheumatoid arthritis [134], multiple sclerosis [135], and chronic obstructive pulmonary disease (COPD), [136] to name a few.

Having described the main PCD mechanisms implicated in *P. aeruginosa* cytotoxins, we now focus on various cytotoxins in *P. aeruginosa* and their modes of cytotoxicity.

## 2. Apoptosis-Inducing Cytotoxins in *Pseudomonas aeruginosa*

*P. aeruginosa* possesses many cytotoxins that induce various forms of apoptotic programmed cell death. Although, it is not clear why *P. aeruginosa* prefers to kill its target host cells by apoptosis, it can be postulated that apoptosis would benefit *P. aeruginosa* during infection in the host because of its anti-inflammatory and immunosuppressive nature [137,138,139]. Below, we will discuss the apoptosis-inducing cytotoxins in *P. aeruginosa* cytotoxins and their mechanisms of action.

### 2.1. Toxin A (ToxA)

Toxin A (ToxA)–A.K.A., Exotoxin A (ExoA), or Pseudomonas Exotoxin (PE)—is an AB toxin secreted by the T2SS [140,141]. AB toxins are composed of A and B components where the A component encodes the active enzymatic domain and the B component is responsible for the transport of the A component across the cytoplasmic membrane of target host cells [142]. Once internalized in host cells, ToxA ADP-ribosylates eukaryotic elongation factor-2 (eEF-2) resulting in the inhibition of protein synthesis and causing apoptotic cell death [143]. ToxA-induced apoptosis exhibits features of both intrinsic and extrinsic apoptosis [144,145]. In mouse embryonic fibroblasts, ToxA induces intrinsic (mitochondrial) apoptosis manifested by rapid degradation of Mcl-1 pro-survival protein and loss of mitochondrial membrane potential [144]. ToxA-induced apoptosis in this cell line was shown to be dependent on BAK (not BAX) oligomerization in the mitochondrial outer membrane, and ToxA-induced apoptosis was completely abolished in cells where Mcl-1 or Bcl-XL were overexpressed [144]. In human mast cells, ToxA induces extrinsic apoptosis [145]. ToxA-intoxicated mast cells manifest extrinsic apoptosis features including activation of initiator Caspase-8 and down-regulation of FLIPs (FLICE-like inhibitory proteins) [145]. Moreover, ToxA-induced apoptosis in this cell line was shown to be dependent on Caspase-8 and Caspase-3 [145]. ToxA deficient strains were shown to be significantly less virulent than the wild-type strain in a mouse model of infection [146]. Although ToxA-induced apoptosis would be expected to be anti-inflammatory, ToxA impacts on immune responses have not been directly investigated. In one study involving the keratitis model of infection, it was shown that ToxA deficient mutant bacteria were rapidly cleared from the eye, with a reduced sign of inflammation at the site of infection [147]. Whether reduced inflammatory responses in the eye were due to reduced bacterial burden or the absence of ToxA remains unknown. Due to its potent cytotoxicity, ToxA has also been extensively evaluated as a potential anti-cancer immunotoxin therapy [148,149].

### 2.2. N-3-Oxododecanoyl Homoserine Lactone (C_12_-HSL)

*N*-3-oxododecanoyl homoserine lactone (C_12_-HSL) is a pheromone that functions as the autoinducer for the Las quorum-sensing in *P. aeruginosa* [150]. In addition to its role in the Las quorum-sensing, C_12_-HSL has been shown to induce various forms of apoptosis, depending on the cell line studied. For example, exposure to C_12_-HSL has been shown to result in the activation of initiator Caspase-8 and the effector Caspase-3 in macrophages and neutrophils [151]. In Jurkat T lymphocytes, C_12_-HSL causes mitochondrial outer membrane damage leading to intrinsic (mitochondrial) apoptosis, mediated by the initiator Caspase-9 [152]. Overexpression of mitochondrial membrane stabilizer Bcl-2 completely abrogated C_12_-HSL-induced apoptosis in Jurkat T lymphocytes [152]. C_12_-HSL has also been shown to induce intrinsic apoptosis by downregulating the STAT3 survival/proliferation pathway in breast carcinoma cells [153]. STAT3 is a known anti-apoptosis transcription factor that protects against intrinsic apoptosis by increasing the expression of anti-apoptotic proteins (i.e., Bcl-2 and Bcl-xL) which stabilize the mitochondrial outer membrane [154]. Similarly, C_12_-HSL causes mitochondrial dysfunction and induces intrinsic apoptosis by attenuating the expression of PGC-1α and its downstream effector BEAS-2B in primary lung epithelial cells [155]. PGC-1α is a master regulator of mitochondrial biogenesis and cellular respiration [156]. Yet, in airway epithelial cells, exposure to C_12_-HSL has been shown to lead to both intrinsic apoptosis—manifested by cytochrome *c* release and Caspase-9 activation—and extrinsic apoptosis, as shown by Caspase-8 activation [157]. Interestingly, *N*-butyryl-L-homoserine lactone (C_4_-HSL), (a closely related autoinducer that activates the Rhl quorum sensing in *P. aeruginosa* [150]), lacks the ability to induce apoptosis likely because of its relatively shorter fatty acid chain [151,152]. Interestingly, at sub-lethal doses C_12_-HSL has been shown to dampen NF-κB activation and suppress TNF-α and IL-12 pro-inflammatory cytokine expression in the RAW264.7 mouse macrophage cell line by triggering the unfolded protein response (UPR) [158]. In contrast, C_12_-HSL has also been shown to induce IL-8 pro-inflammatory cytokine production in human epithelial and fibroblast cells through the activation of NF-κB and AP-2 [159]. Moreover, C_12_-HSL injection into the skin of mice led to the production of inflammatory mediators (IL-1α, IL-6, and MIP-2) in vivo [160].

### 2.3. Azurin

Azurin is a cupredoxin-type electron transfer protein that is involved in electron transfer during denitrification in *P. aeruginosa* [161]. Azurin has also been shown to induce apoptosis in J774 macrophages and many cancer cell lines by stabilizing tumor suppressor p53 and enhancing its activity, which in turn leads to increased levels of ROS and cell demise [162,163,164]. Notably, P53 has been shown to induce intrinsic apoptosis, involving mitochondrial outer membrane damage, cytochrome *c* release into the cytosol, and activation of Caspase-9 through the Apoptosome [165]. It also targets the non-receptor tyrosine kinases (NRTKs) signaling network [166]. Azurin expression has been shown to be elevated in *P. aeruginosa* isolates of the lung in CF patients [167]. This finding likely reflects the need for azurin’s redox function in sustaining *P. aeruginosa* within the anaerobic or microaerobic environment in CF airways [167,168]. Whether azurin’s function as a cytotoxin plays any physiological role for *P. aeruginosa* in vivo remains to be determined.

### 2.4. Pyocyanin

Pyocyanin is another important virulence factor secreted by the T2SS [169,170,171]. Pyocyanin is a water-soluble blue–green, non-fluorescent phenazine-derived pigment metabolite that is capable of oxidizing and reducing molecules and generating reactive oxygen species (ROS) [172,173]. In the environment, pyocyanin has been shown to have bactericidal activity against many bacteria, particularly Gram-positive bacteria such as *Staphylococcus aureus* [174]. The mechanism underlying pyocyanin-induced cytotoxicity in *S. aureus* was shown to involve ROS production [175]. Pyocyanin has also been shown to induce apoptosis in neutrophils by mitochondrial damage and increased ROS [176,177,178,179]. Pyocyanin also impacts immune responses in the host. In mammalian hosts, pyocyanin exposure leads to increased IL-8 via activation of MAPK and NF-kB pathways [180,181]. Despite this pro-inflammatory effect, which could be counterproductive to pathogenicity, pyocyanin plays an important virulence function for *P. aeruginosa* in vivo. Pyocyanin has been shown to be crucial for *P. aeruginosa* in establishing chronic infection and in inducing lung damage in mice [182,183]. Consistent with these reports, pyocyanin is detected in large quantities in the sputum of cystic fibrosis (CF) patients infected with *P. aeruginosa*, enough to inhibit ciliary beating and cause toxicity in the respiratory epithelium in vitro [184].

There are also other apoptosis-inducing cytotoxins that are discussed below under the categories of membrane-associated cytotoxins and Type III Secretion System (T3SS or TTSS) Exotoxins.

## 3. Membrane-Associated Cytotoxins

### 3.1. Lipopolysaccharide (LPS)

Lipopolysaccharide (LPS) is another cell-bound major virulence factor of *P. aeruginosa* [185,186]. LPS is found in the outer membrane of the bacterium and is composed of a hydrophobic domain called Lipid A which is anchored onto a core polysaccharide and a hydrophilic tail made of O-specific polysaccharide [185]. The O-specific polysaccharide is variable and is useful for serotyping strains of *P. aeruginosa* [187]. LPS is an integral part of the outer membrane of Gram-negative bacteria, providing structural support for the bacteria, serving as an adhesin, and well as protection from the environment [185,188]. LPS has also been shown to contribute to *P. aeruginosa* pathogenesis by facilitating host cell adhesion through binding to the ganglioside asialo-GM1 found on epithelial cells [189]. LPS recognition by the NLRC4 (a.k.a. IPAF) canonical or Caspase-11 non-canonical inflammasomes has also been shown to result in Caspase-1 or Caspase-11-dependent pyroptotic cell death [190,191]. In addition, LPS has also been shown to induce apoptosis in A549-transformed lung cells by enhancing ROS production via downregulation of the anti-apoptotic Sirtuin1 (SIRT1) [192].

As for its impact on inflammatory responses, LPS is massively immunogenic and can trigger inflammatory responses through TLR4 recognition [193]. LPS-triggered TLR4 signaling results in the production of pro-inflammatory cytokines which also contribute to the pathology associated with both sepsis and bacteremia [194,195,196]. In addition, cytoplasmic recognition of LPS can trigger Caspase-11-dependent pyroptotic cell death which is highly pro-inflammatory in nature [197]. Given the immunogenicity of LPS, it has been a target for developing anti *P. aeruginosa* vaccines; however, there has been limited success thus far due to the variability of the O-specific polysaccharide [198].

### 3.2. Flagella

Flagella are membrane associated appendages that perform many virulence functions for *P. aeruginosa*. Flagella mediate adhesion to biotic and abiotic surfaces [199,200], mediate swimming motility [201] and function in biofilm formation and maturation [202,203]. Numerous animal models have shown that motility enhances bacterial dissemination and virulence in the host. For example, in neonate mice, flagellation has been demonstrated to enhance virulence in bacteremia and pneumonia models of infection [204]. Similarly, flagellated strains have been shown to cause more damage and exacerbate burn wounds than non-flagellated isogenic strains [205,206]. In addition, *P. aeruginosa* flagellin recognition by NLRC4/IPAF inflammasome can trigger Caspas-1 dependent pyroptosis [190].

Having flagella is beneficial in increasing dissemination and enhancing infection; however, flagella can also be a source of vulnerability for *P. aeruginosa*, as monomeric flagellin detection by Toll-like receptor 5 (TLR5) has been shown to trigger robust inflammatory responses in various immune leukocytes, resulting in diminished bacterial survival in an acute lung infection model [207,208]. In addition, cytoplasmic flagellin recognition by Naip5 and Naip6 [209,210] can also lead to activation of NLRC4 canonical inflammasome which further amplifies inflammatory responses against *P. aeruginosa* infection [211]. As a result, *P. aeruginosa* strains often downregulate the expression of flagellar components after the establishment of infection and during chronic infection to evade host innate immune responses [212,213,214]. Downregulation of immunogenic virulence factors and/or structures (i.e, flagellum) is a common theme among *P. aeruginosa* infections, especially in CF patients, where expression of virulence factors could hinder bacterial survival rather than aid in its dissemination [215,216].

### 3.3. Porins

*P. aeruginosa* also possesses over 20 porins in its outer membrane that serve many crucial physiological functions essential for virulence including nutrient uptake, adhesion, decreasing permeability to antibiotics, and signaling ([217,218]). For example, OprF, one of the major porins in the *P. aeruginosa* outer membrane, has been shown to be required for full virulence of *P. aeruginosa* [219]. Deletion in the *oprF* gene impairs many virulence-associated functions including colonization, quorum sensing (QS), and secretion through the T3SS [219]. In addition to these virulence functions, purified porins have also been shown to induce intrinsic apoptosis in epithelial target host cells, as manifested by reductions in the *bcl-2* gene expression [220].

As crucial as they are to the pathophysiology of *P. aeruginosa*, porin recognition by pattern recognition receptors (PRRs) can also lead to the production of pro-inflammatory cytokines, inflammatory responses, and complement activation, thus interfering with *P. aeruginosa*’s ability to colonize and cause infection [221,222]. Interestingly, adoptive transfer of dendritic cells immunized with wild-type or recombinant OprF ex vivo has been shown to be protective against *P. aeruginosa* lung infection in mice [223].

### 3.4. Rhamnolipids

*P. aeruginosa* can produce about 30 different congeners of surface-active rhamnolipids, which are glycolipid biosurfactants composed of mono- or di-rhamnose linked to 3-hydroxy-fatty acids of different lengths [224,225,226]. Rhamnolipids have been detected in large quantities (range: 8–65 µg/mL) in the sputum of CF patients, and their presence has been associated with CF lung pathology [227,228]. Because of their high potential for use in various biotechnological applications *P. aeruginosa* rhamnolipids have been investigated extensively [226,229,230].

Rhamnolipids play several important virulence functions for *P. aeruginosa*. First, rhamnolipid-expressing *P. aeruginosa* strains, as well as purified rhamnolipids, have been shown to compromise the barrier function of human respiratory epithelium by disrupting the tight junctions, thus facilitating the paracellular invasion by *P. aeruginosa* [231]. Second, exposure to rhamnolipids has been shown to interfere with ciliary beating and mucociliary clearance of *P. aeruginosa* in tracheal rings of guinea pig animal models [227]. Third, rhamnolipids have also been shown to aid *P. aeruginosa* in swarming motility [232]; in turn, swarming motility has been demonstrated to regulate the expression of virulence genes and antibiotic resistance in *P. aeruginosa* [233]. Forth, rhamnolipids have also been demonstrated to play a role in structural biofilm development and in maintaining channels between multicellular structures in biofilms [234,235]. Fifth, rhamnolipids appear to modulate *P. aeruginosa* membrane composition by reducing LPS and porin membrane proteins [236], thus potentially contributing to their well-known intrinsic resistance towards antibiotics [237]. Rhamnolipids can also induce cytotoxicity in target epithelial cells. MCF7 breast cancer cells intoxicated with mono- and di-rhamnolipids from *P. aeruginosa* undergo apoptotic cell death, manifested by nuclear condensation and fragmentation, p53 activation, mitochondrial damage, and appearance of sub-G1 (apoptotic) subpopulations [238,239]. Rhamnolipids have been shown to cause potent necrotic cell death in human and murine neutrophils and macrophages [240,241].

As is the case for many virulence factors and virulence structures in *P. aeruginosa*, rhamnolipids can both trigger and combat host innate immune responses. For instance, rhamnolipids are required for the expression of flagellin-induced psoriasin (S100A7) antimicrobial peptides and chemokines in human skin [242]. In this context, the production of rhamnolipids may be detrimental to *P. aeruginosa* pathogenesis in vivo. In contrast, rhamnolipids have been shown to cause potent necrotic cell death in humans and murine neutrophils and macrophages, therefore protecting *P. aeruginosa* against clearance in the lungs [240,241].

## 4. Type III Secretion System (T3SS or TTSS) Exotoxins

T3SS is a highly conserved virulence structure that is found in several other important pathogenic Gram-negative bacteria, including *P. aeruginosa* [243,244,245]. T3SS arguably plays the most significant role in the pathogenesis of T3SS-expressing *P. aeruginosa* and without it, T3SS-expressing *P. aeruginosa* becomes severely attenuated in its ability to cause infection [243,246,247]. Six effector proteins, referred to as exotoxins (Exo) have been shown to be secreted through the T3SS of *P. aeruginosa* but only four of these effector proteins (ExoS, ExoU, ExoY, and ExoT) have been demonstrated to possess virulence functions, including cytotoxicity [243,248]. Due to the narrow channel of the needle complex, the effector proteins must be secreted in their unfolded state [249], and because of this, they all require specific chaperones which can help maintain the effector toxins in an unfolded state while in the bacterial cytosol and during translocation and can also aid targeting the effector to the T3SS apparatus [250,251,252]. In addition, all exotoxins require host cofactors for their activation within the host cell [253,254,255]. The dependence of activity on host cofactors ensures that *P. aeruginosa* is protected from the harmful activities of these virulence factors. We will now discuss the T3SS and its effector proteins with demonstrated virulence functions including cytotoxicity.

### 4.1. T3SS Apparatus

A growing body of data has demonstrated that insertion of the T3SS apparatus of Gram-negative pathogens, including *P. aeruginosa* in the host plasma membrane results in cell death [256,257,258]. It is assumed that the damage caused by the T3SS pore-forming activity is the cause of passive necrotic cell death due to trauma and membrane leakage. However, the T3SS-induced necrosis has been shown to be completely blocked by ExoT [259], suggesting that the T3SS-induced cell death is a form of programmed cell death and not accidental necrosis, occurring as a consequence of T3SS-induced massive trauma to the membrane. Similarly, the T3SS in *Yersinia* has also been shown to cause pore formation and induce cytotoxicity in target host cells and the *Yersinia* T3SS-induced cytotoxicity was also shown to be blocked by YopE effector toxin [257], which is a homolog of the GAP domain of ExoT [260,261]. Two recent reports have indicated that the cell death induced by the T3SS of *P. aeruginosa* is pyroptosis [262,263]. As discussed above, pyroptosis is mediated by Caspases 1 and 11 (in mice) and 4 and 5 (in human), which are inhibited by the pancaspase inhibitor z-VAD. However, the T3SS-induced cytotoxicity was not appreciably affected by z-VAD [259], suggesting that T3SS-induced cytotoxicity is not pyroptosis. Given that the T3SS-induced cytotoxicity is completely abrogated by a toxin (ExoT) that induces potent apoptosis (discussed below) suggests that the T3SS-induced cytotoxicity is necroptosis because it is prevented under apoptotic conditions, as discussed above. Whether necroptosis and/or pyroptosis is the underlying mechanism of T3SS-induced cytotoxicity remains to be further investigated.

As for the impact of the T3SS on the host’s innate immune responses, this virulence structure is perhaps the most prominent trigger of inflammatory responses in the host during infection. Various inflammasome subtypes (e.g., NLRP3 and/or NLRC4) have been implicated in the recognition of and in responses to T3SS and *P. aeruginosa* infection, although NLRC4 canonical inflammasome appears to be the primary mode of T3SS recognition in BMDMs and in host tissues [19,190,264,265,266,267]. There also appears to be some contradictory reports regarding the impact of T3SS-triggered inflammatory responses on the outcome of infection, in that the same inflammasome (NLRC4) has been shown to be either crucial in *P. aeruginosa* clearance, thus benefiting the host; or paradoxically facilitating bacterial colonization and enhancing *P. aeruginosa* pathogenesis, thus benefiting the pathogen. For example, Franchi et al. demonstrated that recognition of T3SS-expressing *P. aeruginosa* by NLRC4 inflammasome triggers the production of IL-1β in intestinal phagocytes that are crucial in limiting *P. aeruginosa* gastric infections [268]. Similarly, NLRC4 was found to contribute to the recognition and clearance of T3SS-expressing *P. aeruginosa* in a wound model [19], and in a cystic fibrosis (CF) lung model of infection [269]. In contrast, Faur et al. demonstrated that Nlrc4-deficient mice showed enhanced bacterial clearance and decreased lung injury contributing to increased animal survival following pleural infection with a T3SS-expressing *P. aeruginosa* strain [270]. These reports suggest that specific organs and/or sites within a host may have evolved distinct mechanism(s) to detect and respond to T3SS and its effectors during *P. aeruginosa* infection.

### 4.2. ExoS

ExoS is a bifunctional protein consisting of an N-terminal GTPase Activating Protein (GAP) domain and a C-terminal ADP-ribosyltransferase (ADPRT) domain that is directly translocated into host cytoplasm through the T3SS [271] using SpcS chaperone protein [272]. Upstream of the GAP domain is a membrane localization domain (MLD) which targets the toxin to the mammalian cytoplasmic membrane [273]. Deletion of the MLD was found to not affect ExoS translocation; however, ADP-ribosylation of Ras, a known target of ExoS, was lost, thus demonstrating the importance of this sequence in mediating ExoS interaction with some of its targets [274].

The GAP domain of ExoS targets RhoA, Rac1, and CDC42 [275,276]. These small Ras-like GTPases are active when bound to GTP and inactive when bound to GDP [277]. ExoS inactivates RhoA, Rac1, and Cdc42 through allosteric interaction of a conserved arginine-finger in its GAP domain with the aforementioned GTPases, stimulating them to hydrolyze their bound GTP to GDP [275,276,277,278]. These small GTPases play important roles in coordinating and maintaining the actin cytoskeleton; thus, inactivating their signaling affects processes, such as cell migration and cell division and leads to cell rounding [279,280].

The ADPRT domain of ExoS has many targets in mammalian cells and requires the host 14-3-3 protein as the cofactor for its activity within host cells [254,281]. Targets include Ras, Rab and Rho family of proteins, as well as ezrin, radixin, meosin, vimentin, and cychlophilin A [282]. ADP ribosylation of these proteins by ExoS/ADPRT can lead to disruption of the cytoskeleton, endocytosis, and cell–cell binding, as well as inhibition of DNA synthesis and ultimately apoptotic cell death [279,282,283,284,285].

ExoS-intoxicated cells display signs of both Caspase-9 dependent intrinsic apoptosis and death receptor-mediated Caspase-8 dependent extrinsic apoptosis and both domains of ExoS contribute to ExoS-induced apoptosis [285,286,287,288,289]. Intoxication with ExoS/GAP has been shown to lead to enrichment of Bax and Bim into the mitochondrial outer membrane; disruption of mitochondrial membrane and release of cytochrome *c* into the cytosol; and activation of initiator Caspase-9 and executioner Caspase-3 caspases, leading to intrinsic/mitochondrial apoptosis in target host cell [285]. ExoS/ADPRT intoxication has been shown to result in the activation of initiator Caspase-8 in a manner that is dependent on the FADD adaptor protein, although ExoS-induced apoptosis was independent of Fas death receptor and Caspase-8 activities [290].

As for ExoS’s impact on host immune responses, ExoS has been shown to either dampen or trigger immune responses during infection. Intoxication of PBMCs, monocytes, and T cells, with ExoS, or recombinant ExoS (rExoS), strongly induced transcription of pro-inflammatory cytokines and chemokines, namely, IL-1α, IL-1β, IL-6, IL-8, MIP-1α, MIP-1β, MCP-1, RANTES [291,292]. The same group further showed that the induction of pro-inflammatory cytokines in ExoS-treated monocytes was due to the activation of TLR4 signaling by the ExoS/GAP domain and TLR2 signaling by ExoS/ADPRT domain activities [293]. Adding to the confusion in the field, Galle et al. reported that ExoS dampened Caspase-1 mediated IL-1β production in macrophages in a manner that was dependent on its ADPRT domain activity [294]. More recently, it was reported that ExoS had no impact on inflammatory responses in a wound model for infection with *P. aeruginosa*, although ExoS was required for full colonization of bacteria in the wound [19]. These discrepancies are likely due to technical differences in these studies.

### 4.3. ExoT

ExoT is the only T3SS effector that is expressed in all T3SS-expressing *P. aeruginosa* clinical strains [295], indicating a more fundamental role for this virulence factor in *P. aeruginosa* pathogenesis. The importance of ExoT to *P. aeruginosa* pathogenesis is further highlighted by the observation that it is actively targeted for degradation by host defenses in epithelial cells. ExoT becomes complexed with Crk and a Crk binding partner Cbl-b, an E3 ubiquitin ligase [296]. As a result, ExoT becomes polyubiquitinated and is targeted for proteasomal degradation. In their study, Balachandran et al. showed that mice lacking Cbl-b were significantly more susceptible to infection by strains expressing ExoT.

ExoT shares 76% protein homology with ExoS and possesses an N-terminal GAP domain and a C-terminal ADPRT domain [271,297,298]. ExoT also contains sequence homology with the MLD of ExoS, although this sequence has not been experimentally mapped [282]. Nevertheless, support for ExoT MLD is demonstrated by similar intracellular fractionation patterns between ExoS and ExoT [278] as well as experiments showing how ADPRT domain switching in ExoS and ExoT maintains their substrate specificities in both chimeras [299].

Similar to ExoS/GAP, the ExoT/GAP domain also targets RhoA, Rac1, and CDC42 [279,280]. In contrast to the ExoS/ADPRT domain, the ADPRT domain of ExoT targets only three non-overlapping substrates; namely, CrkI and CrklI isoforms of Crk adapter protein and phosphoglycerate kinase 1 (PGK1) glycolytic enzyme [300]. Similar to ExoS, the ADPRT domain of ExoT also requires the host 14-3-3 protein as the cofactor for its activity [254]. ExoT has been shown to inhibit bacterial phagocytosis by macrophages, cell migration, and cause cell rounding in a manner that is primarily dependent on its GAP domain activity, although the ADPRT domain also contributes [21,298,300,301,302]. ExoT also exerts potent anti-proliferative effects in its target host cells [303]. The GAP domain of ExoT has been shown to inhibit cell division in epithelial cells by inhibiting the early stage of cytokinesis at the cleavage furrowing step, likely through its inhibitory effect on RhoA; whereas the ADPRT domain blocks the late stage of cytokinesis at the abscission step by targeting CrkI [303]. In addition, both domains of ExoT have been shown to cause cell cycle arrest in G1 interphase in melanoma cells by dampening the expression of G1/S checkpoint proteins ERK1/2, cyclin D1, and cyclin E1 [304].

ExoT is also a potent inducer of apoptotic cell death in its target hosts and both domains contribute to this virulence activity [259,305]. The ExoT/ADPRT was shown to be necessary and sufficient to induce anoikis apoptosis by transforming Crk adaptor protein into a cytotoxin which interfered with the integrin survival signaling by destabilizing the focal adhesion sites through persistent activation of the anoikis mediator, p38β [306]. The ExoT/GAP was shown to be necessary and sufficient to induce intrinsic/mitochondrial apoptosis by activating the initiator Caspase-9 and the effector Caspase-3 through upregulation of the expression and subcellular mobilization of Bax, Bid, and Bim—pro-apoptotic Bcl2 family of proteins—into mitochondrial outer membrane [307]. Interestingly, the ExoT/ADPRT-induced anoikis apoptosis has faster kinetics occurring within 5.5 ± 1.3 h, whereas the ExoT/GAP-induced mitochondrial apoptosis shows slower kinetics occurring within 16.2 ± 1.3 h in intoxicated cells [306,307].

It is important to note that while pre-treatment with the pancaspase inhibitor z-VAD effectively protects eukaryotic cells from ExoT-induced apoptosis, it does not protect the host cells from ExoT-induced disruption of actin cytoskeleton [307], ExoT-induced focal adhesion site disassembly [306], or ExoT-mediated anti-proliferative effects on cytokinesis [303], or ExoT-mediated induction of G1 cell cycle arrest in target host cells [304], indicating that ExoT-induced apoptosis can be uncoupled from ExoT’s other virulence functions.

As for ExoT’s impact on host immune responses, Mohamed et al. recently demonstrated that ExoT inhibits IL-1β and IL-18 pro-inflammatory cytokines production in primary macrophages by inhibiting the phosphorylation cascade through Abl→PKCδ→NLRC4 by targeting CrkII, which they further showed to be required for Abl transactivation and NLRC4 canonical inflammasome activation in response to T3SS and *P. aeruginosa* infection [19]. They corroborated these in vitro data in an animal model of wound infection, showing that recognition of T3SS leads to the phosphorylation cascade through Abl→PKCδ→NLRC4, culminating in the activation of NLRC4 inflammasome in response to *P. aeruginosa* infection. Interestingly, they showed that in the wound infection model, ExoT was the primary anti-inflammatory agent for *P. aeruginosa,* and other T3SS effector proteins (ExoU and ExoS) had no impact on inflammatory responses in wound tissues [19].

### 4.4. ExoU

ExoU is a potent inducer of rapid necrotic cytotoxicity in target eukaryotic host cells [243,255,308,309]. ExoU has a patatin-like domain that contains phospholipase A_2_ activity and can target phospholipids, lysophospholipids, and neutral lipids [255,308,310]. ExoU utilizes the chaperone protein called SpcU for secretion through the T3SS [311]. ExoU also requires host DNAJC5 chaperone and ubiquitin as the cofactor for its activity within the target host cell [253,254,255,312,313,314]. The MLD (membrane localization domain) of ExoU has been mapped to residues 550–687 in its C-terminal domain [315]. This allows ExoU to target the plasma membrane where it can carry out its phospholipase activity [316].

The necrotic nature of ExoU-induced cytotoxicity would suggest a pro-inflammatory consequence for this toxin in the host environment. Consistent with this notion, excessive inflammatory responses due to ExoU-induced endothelial barrier disruption have been shown to culminate in the acute respiratory distress syndrome (ARDs) in a pneumonia animal model of infection [317]. Intriguingly, ExoU has also been shown to function as an anti-inflammatory agent for *P. aeruginosa*. In a pneumonia model of infection, ExoU was shown to create a localized immunosuppressed zone in the vicinity of bacteria by directly killing phagocytic leukocytes (neutrophils and macrophages), albeit there were more inflammatory mediators in the lungs of mice infected with ExoU-expressing *P. aeruginosa* strain [318,319]. In another report, ExoU was shown to dampen IL-1β pro-inflammatory cytokine production by inhibiting Caspase-1-dependent NLRC4 (a.k.a., IPAF) activation in macrophages [320]. In the same report, ExoU was shown to reduce serum IL-1β and enhance bacterial fitness in a systemic model of infection in mice. To add to the confusion, in a wound model of infection in mice, it was recently demonstrated that ExoU had no impact on pro-inflammatory cytokines production and inflammatory leukocyte responses in a murine wound model of infection [19].

### 4.5. ExoY

ExoY is an adenylyl and guanylyl cyclase that shares sequence homology with *Bordetella pertussis* CyaA and *Bacillus anthracis* edema factor [321,322]. In a recent study, ExoY was detected in 93% of clinical isolates in critically ill pneumonia patients who tested positive for *P. aeruginosa,* and its presence was associated with end-organ dysfunction in this patient cohort [323]. ExoY requires binding to filamentous actin (F-actin) for its activity [324]. The primary activity of ExoY on mammalian cells appears to be as an edema factor, increasing vascular permeability [322,325]. However, ExoY possesses other virulence activities including disruption of actin cytoskeleton [321,326], inhibition of phagocytic uptake by the host immune system [327], and inhibition of endothelial repair after injury [328]. ExoY drives vascular permeability through its adenylyl cyclase activity which causes Tau hyperphosphorylation and insolubility [322].

In one report, ExoY was also shown to induce cell lysis in Madin–Darby canine kidney (MDCK) epithelial cells, as determined by the release of lactate dehydrogenase (LDH) into the culture supernatant [329]. In another report, infection with ExoY-expressing *P. aeruginosa* was associated with increased apoptosis in the lung of infected mice [330]. However, in a recent report, ExoY was found to cause an accumulation of active Caspase-7 without causing cell death in pulmonary microvascular endothelial cells (PMVECs). Whether or not ExoY is a bona fide cytotoxin requires further investigation.

As for ExoY’s impact on host immune responses, Kloth et al. recently reported that infection with ExoY-expressing *P. aeruginosa* was associated with elevated levels of pro-inflammatory cytokines in the sera and the bronchoalveolar lavage fluids (BALFs) and increased infiltration of neutrophilic granulocytes in the perivascular space in an acute airway infection model in mice [330]. Interestingly, these effects were also observed when ExoY was catalytically inactive, suggesting that at least initial inflammatory responses to ExoY are independent of its catalytic activity. Since in these studies, the control infections with the ExoY-deficient and the T3SS mutant strains were not included, it remains unclear whether these pro-inflammatory responses in host tissue were directed at ExoY or whether they were in response to the functional T3SS which itself is a potent inducer of inflammatory responses [19].

## 5. Other Pore-Forming Cytotoxins

We discussed the T3SS-mediated pore-forming cytotoxicity above. Here, we discuss other known pore-forming cytotoxins in *P. aeruginosa*.

### 5.1. CTX

CTX is a cytotoxin contained within the øCTX pro-phage in lysogenized *P. aeruginosa* bacterial strains such as PA158 [331,332]. CTX is expressed as a pro-cytotoxin of 286 amino acids, with both its activation and secretion requiring the removal of 20 amino acids from its C-terminus [333]. CTX has been shown to cause cytotoxicity in leukocytes through its pore-forming activity [332,334]. CTX inactivation has been demonstrated to result in reduced virulence in a systemic infection model in leukopenic mice, indicating that it is required for full *P. aeruginosa* pathogenesis in this model [335]. The impact of CTX cytotoxicity on immune responses has not been directly investigated. However, given that killing by pore formation would result in the release of DAMPs [336], CTX-mediated cytotoxicity is likely pro-inflammatory in nature.

### 5.2. Exolysin (ExlA)

Phylogenetic analyses based on whole-genome sequencing of various *P. aeruginosa* strains have recently revealed a new clade (PA7-like clade) of highly cytotoxic *P. aeruginosa* strains that lack the genes for the T3SS and its effectors [337,338,339]. The cytotoxicity associated with the PA7-like clade is attributed to ExolysinA (ExlA) cytotoxin which causes potent cytotoxicity in eukaryotic cells by pore formation [339,340]. ExlA export across the *P. aeruginosa* membrane is dependent on an outer membrane protein, ExlB, and together ExlA and ExlB define a new active two-partner secretion (TPS) system in *P. aeruginosa* [339,340]. ExlA/B-expressing *P. aeruginosa* strains are highly virulent in mice, causing lung hemorrhage and septicemia in a manner that is dependent on the ExlA/B expression [339]. ExlA was also found to be essential for bacterial dissemination to the liver and spleen [339]. These strains have also been known to cause necrotic lesions in pneumocytes and endothelial cells, resulting in alveolo-vascular barrier breakdown in mice [341]. Interestingly, exposure to ExlA results in Caspase-1/Caspase-11--dependent cytotoxicity in BMDMs [342], suggesting that pyroptosis may be the underlying mechanism of ExlA-induced cytotoxicity [51]. However, the same group also reported that ExlA-induced cytotoxicity in BMDMs is also blocked by RIP1 kinase inhibitors, necrostatin-1, and necrostatin-5 [342], suggesting that necroptosis may also be involved [343,344]. Clearly, more work is needed to elucidate the mechanisms underlying ExlA-induced cytotoxicity in various cell lines and in vivo.

Regardless, ExlA cytotoxicity has been shown to be highly inflammatory in nature. ExlA has been demonstrated to promote the maturation of IL-1β through the NLRP3 inflammasome leading to alveolar tissue damage and inflammation in the lungs of mice infected with ExlA-expressing *P. aeruginosa* [341]. The authors further showed that deficiency in Caspase-1/11 protected against alveolar tissue damage and improve survival in infected mice, by partially reducing the inflammatory environment. Interestingly, the bacterial burden was significantly lower in the lungs of Capase-1/11 knockout mice as compared to wild-type mice, indicating that Caspase-1/11 function may enhance bacterial fitness in this environment and at this time point. Even more surprisingly, the levels of IL-1β and IL-18 were similarly elevated in the lungs of wild-type and Caspase-1/11 knockout mice infected with ExlA-proficient strain, suggesting an inflammasome-independent mechanism for IL-1β and IL-18 processing and activation under these conditions. It is worth noting that despite higher initial inoculum, the infection titers of the ExlA-expressing *P. aeruginosa* strain were 3–4 log order lower than the T3SS-expressing *P. aeruginosa* strain that was used in these studies, suggesting a fitness disadvantage for the PA7-like clade in the mammalian host.

## 6. Cytotoxins as Potential Therapeutic Targets

*Pseudomonas aeruginosa* infections are very difficult to treat because this pathogen has evolved a plethora of intrinsic and acquired resistance mechanisms to current antibiotics [345,346,347,348,349,350]. Since *P. aeruginosa* cytotoxins play crucial roles in its pathogenesis in vivo, targeting these toxins, (as novel strategies to reduce *P. aeruginosa* virulence and render this pathogen incapable of causing the disease), has been gaining momentum. For example, T3SS and its effectors play a pivotal role in *P. aeruginosa* pathogenesis and without it, *P. aeruginosa* becomes avirulent and cannot cause disease [20,243,246]. In a recent report, the T3SS inhibitors (salicylidene acylhydrazide INP0341 and of hydroxyquinoline INP1750) were shown to reduce cell death and inflammasome activation by clinical isolates expressing T3SS toxins [351]. INP0341 has also been shown to prevent corneal infection by *P. aeruginosa* in an experimental model of murine keratitis [352]. Other strategies to block the T3SS (e.g., anti-PcrV antibody, phenylacetic acid), or inhibit its effectors, (e.g., ExoU inhibition by Phospholipase A2 inhibitors or ExoS inhibition by small molecules), have also been shown to protect eukaryotic cells and tissues from cytotoxicity and *P. aeruginosa* infection [255,353,354,355]. Similarly, inhibitors of quorum sensing, rhamnolipids, and pyocyanin have also shown promising results in protecting against cellular cytotoxicity and tissue damage during *P. aeruginosa* infection [240,356,357]. Clinical trials are needed to evaluate these strategies as viable therapeutics for *P. aeruginosa* infections.

## 7. Concluding Remarks

*P. aeruginosa* is a highly successful bacterial pathogen capable of infecting numerous hosts, as diverse as plants and mammals [358,359,360]. The induction of cell death in target host cells appears to be a major virulence strategy that is fundamental to *P. aeruginosa* pathogenesis during infection, as manifested by many virulence factors that serve this purpose for *P. aeruginosa*, as discussed in this review and summarized in Table 1. It is unlikely that *P. aeruginosa* evolved all these cytotoxins as redundant means to merely kill its target host cells without consideration for the consequences associated with their different modes of cytotoxicity on the host immune system. It is more likely that depending on host and environmental factors, *P. aeruginosa* may prefer to use a subset of these virulence cytotoxins to fulfill its needs. For example, early after infection where bacteria numbers are few, *P. aeruginosa* is vulnerable, thus it may prefer to deploy apoptotic-inducing cytotoxins as a way to remain stealth, given the anti-inflammatory nature of apoptosis. However, the caveat with apoptotic cell deaths may be the slower kinetics of cell death due to the dependence of apoptosis on the elaborate protein-based complexes mediating this form of cell death as discussed above. Conversely, as the bacteria numbers increase during infection, it is inevitable for the host to recognize and mount effective immune responses against *P. aeruginosa*. Under such conditions, *P. aeruginosa* may prefer to use potent and fast-acting necrotic and pore-forming toxins, such as ExoU, to eliminate the threat posed by host immune phagocytic leukocytes. In addition, it remains unclear which mode of cytotoxicity may prevail in response to these cytotoxins. For example, apoptosis and necroptosis are mutually exclusive, as discussed above, so the mode of cytotoxicity may depend on the level of the competing cytotoxins and their kinetics of cytotoxicity. Moreover, other chemical and/or physical factors, (e.g., aerobic vs. anaerobic environment, tissue vs. blood, presence or absence of cofactors, pH, etc.), may also influence the choice of cytotoxins for *P. aeruginosa*. Clearly, more work is needed to tease out these possibilities.

It is worth noting that the impacts of some of the cytotoxins on host immune responses were at times confusing (in that they had been implicated in both pro-inflammatory or anti-inflammatory responses), or not congruent with the nature of the cell death induced by these toxins as discussed above and summarized in Table 1. These seeming discrepancies may be due to differences in experimental procedures that in most studies primarily were reliant on in vitro cell culture-based assays. Or, they may reflect the possibility that specific sites within a host may have evolved different mechanism(s) to detect and/or respond to *P. aeruginosa* and its cytotoxins during infection. It is therefore prudent to assess immune responses in appropriate in vivo models as a way to evaluate the net impact of *P. aeruginosa* cytotoxins on host immune responses.

## Figures and Tables

**Figure 1 cells-12-00195-f001:**
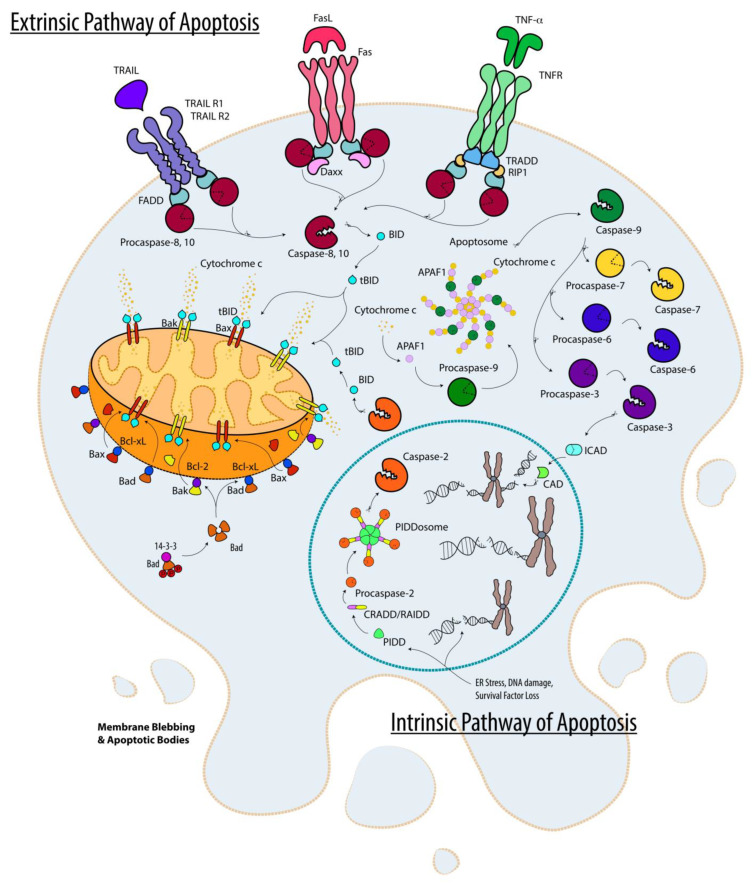
Apoptosis. The intrinsic (mitochondrial) and the extrinsic (death receptor-mediated) pathways of apoptotic cell deaths are depicted. These pathways are described in the text.

**Figure 2 cells-12-00195-f002:**
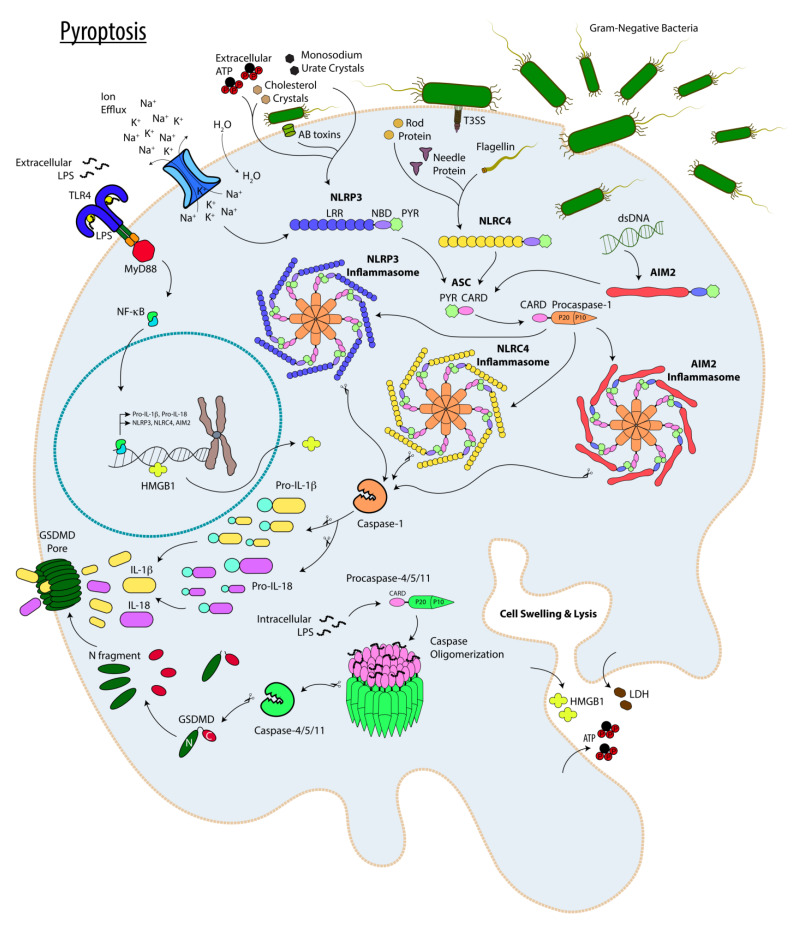
Pyroptosis. The main players involved in pyroptotic programmed cell death are depicted. The pathways are described in the text.

**Figure 3 cells-12-00195-f003:**
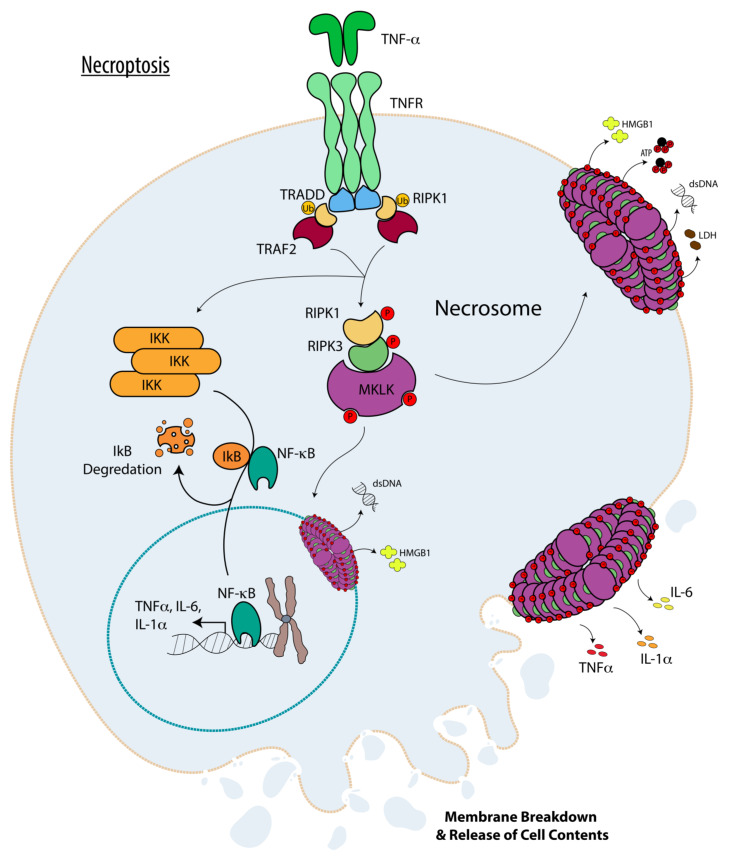
Necroptosis. The main players involved in necroptotic programmed cell death are depicted. The pathways are described in the text.

**Table 1 cells-12-00195-t001:** Summary of the cytotoxic in *P. aeruginosa* strains and their mode of cytotoxicity and impacts on immune responses.

Toxin	Function	Toxin Type	Host Target	Cytotoxicity Type	Host Immune Impact
**ToxA**	ADP-ribosylation [140,141]	AB Toxin [140,141]	eEF-2 [143]	Intrinsic and extrinsic apoptosis [144,145]	Anti-inflammatory [147]
**C_12_-HSL**	Autoinducer for the Rhl qourum-sensing [150]	Apoptotic cytotoxin [153,155]	Dampens expression of STAT3 and BEAS-2B [153,155]	Intrinsic and extrinsic apoptosis [151,152,153]	- Pro-inflammatory [159,160] - Anti-inflammatory [158]
**Azurin**	Cuperedoxin protein; Redox modulation [161]	Apoptotic cytotoxin [162,163,164,166]	p53 and non-receptor tyrosine kinases (NRTKs) [162,163,164,166]	Intrinsic apoptosis [162,163,164,165,166]	Unknown
**Pyocyanin**	- Pigment molecule [169,170,171] - Redox modulation [172,173]	Apoptotic cytotoxin [176,177,178,179]	Causes mitochondrial dysfunction [176,177,178,179]	ROS and intrinsic apoptosis features [176,177,178,179]	Pro-inflammatory [180,181]
**Lipopolysaccharide (LPS)**	Cell wall stabilization and adhesin [185,188]	Pyroptotic cytotoxin and possibly apoptosis [190,191,192]	- TLR4, Caspase-11, and SIRT1 [190,191,192]	- Pyroptotic cell death via Caspases 1 and 11 [190,191] - ROS-induced apoptosis [192]	Highly pro-inflammatory via several mechanisms [193,194,195,196,197]
**Flagella**	Motility, biofilm [201]	Membrane disruption [199,200]	- TLR5 [207,208] - Naip5 and Naip 6 [209,210]	Pyroptotic cell death via Caspase-1 inflammasome [190].	Highly pro-inflammatory via several mechanisms [207,208,209,210,211]
**Porins**	Nutrient uptake; adhesin; signaling [217,218]	Causes mytochondrial dysfunction [220]	Dampens expression of BCL-2 [220]	Intrinsic apoptosis [220]	Pro-inflammatory via activation of PRRs [221,222]
**Rhamnolipids**	Glycolipid biosurfactants; involved in swarming motility, biofilm development, membrane composition, and antibiotic resistance [232,234,235,236,237]	- Apoptotic cytotoxin [238,239] - Necrotic cytotoxin [240,241]	Mitochondrial outer membrane and p53 [238,239].	- Features of intrinsic apoptosis [238,239] - Necrotic cell death feature as well [240,241]	- Pro-inflammatory [240,241,242]
**T3SS**	Translocation machinery essential for *P.aeruginosa* pathogenesis [243,246,247]	- Pyroptotic [262,263] - Potentially necroptotic [181,182,183,184]	Host membrane [256,257,258]	- Pyroptosis [262,263] - Necroptosis [181,182,183,184]	Pro-inflammatory [19,188,189,190,191,192,193,194]
**ExoS**	Bifunctional exotoxin (GAP/ADPRT) toxin involved in many virulence functions [271,275,276,277,278,279,280]	Apoptotic cytotoxin [285,290]	**GAP**: RhoA, Rac1, CDC42 [278,284] **ADPRT**: Ras, Rab, Rho, Ezrin, Radixin, Meosin, Vimentin, Cyclophilin A [279,282,283,284,285]	**GAP**: Intrinsic and extrinsic apoptosis [285] **ADPRT**: Features of extrinsic apoptosis [290]	- Pro-inflammatory [291,292,293] - Anti-inflammatory [294] - No effect on immune responses in wound tissues [19]
**ExoT**	Bifunctional exotoxin (GAP/ADPRT) toxin involved in many virulence functions [21,298,300,301,302,303,304]	Apoptotic cytotoxin [259,305,306,307]	**GAP**: RhoA, Rac1, CDC42 [279,280] **ADPRT**: CrkI, CrkII, PGK1 [300]	**GAP**: Intrinsic apoptosis [259,305,307] **ADPRT**: Anoikis apoptosis [306]	Anti-inflammatory by blocking NLRC4 inflammasome [19]
**ExoU**	Patatin-like phospholipase Exotoxin [255,308,310]	Necrotic cytotoxin [255,308,309,361]	Phospholipids, lysophospholipids, neutral lipids [255,308,310]	Necrosis [180,233,234,235]	- Pro-inflammatory due to necrosis [317,318,319] - Anti-inflammatory [320] - Local immunosuppressant due to killing leukocytes [318,319] - No effect on immune responses in wound tissues [19]
**ExoY**	Adenylyl and guanylyl cyclase Exotoxin involves in several virulence functions [321,322,325,327,328]	Exotoxin	Filamentous actin and Tau [322,324]	Cell lysis [255], apoptosis [256]	- Pro-inflammatory [330] - No effect on immune responses in wound tissues [19]
**CTX**	øCTX pro-phage cytotoxin [331,332]	Pore-forming cytotoxin [332,334]	Host membrane [332,334]	Cell lysis [258,260]	Unknown but likely pro-inflammatory due to release of DAMPs [336]
**ExlA**	AB cytotoxin required for pathogenesis of PA7-like clade of *P. aeruginosa* strains [339]	Pore-forming [339,340]	Host membrane translocation mediated by ExlB [339,340]	- Pyroptosis, mediated by Caspase-1/11 [342] - Necroptosis, mediated by RIP1 kinase [342]	Highly pro-inflammatory [341]

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
