# Peer review of "Pseudomonas aeruginosa Cytotoxins: Mechanisms of Cytotoxicity and Impact on Inflammatory Responses"

_cells, 2023, doi:10.3390/cells12010195_

Round 1

Reviewer 1 Report

Dear Editors of the Cells Journal

I hope you are doing well.

Attached are my recommendations for the submitted manuscript entitled “Pseudomonas aeruginosa Cytotoxins: Mechanisms of Cytotoxicity and Impact on Inflammatory Responses”.

Thanks for providing the opportunity to review the manuscript.

With thanks

Author Response

We thank the reviewer for the thorough review of our manuscript and for his/her insightful comments. We are also very grateful to the reviewer for his/her recognition of the strengths of this review article, calling it “very comprehensive, organised and easy to follow.” Below, please find our point-by-point responses to your queries.

  1. Briefly discuss the impact and specific role of efflux pumps in stimulating cytotoxicity and inflammatory pathways.

Response: Respectfully, we are not aware of any publications on P. aeruginosa efflux pumps being implicated in cytotoxicity. We did another search in Google Scholar & Pubmed and did not find any articles on P. aeruginosa efflux pumps inducing cytotoxicity in target host cells. If there are publications that we overlooked, we apologize to the reviewer and would appreciate it if the reviewer would provide us the PMID of the articles so that we can appropriately acknowledge it in our review manuscript. P. aeruginosa efflux pumps have been shown to play crucial roles in resistance to antibiotics and there are also reports on efflux pumps adversely impacting T3SS expression and activity and P. aeruginosa virulence in vivo. However, since, this review article is primarily focused on cytotoxins in P. aeruginosa, we believe that including such information on efflux pumps would detract from this review article.

  1. The authors need to mention the innovative strategies that may be employed to combat drug resistance in P. aeruginosa.

Response: We respectfully argue that including such information in this review article detracts from its focus which is P. aeruginosa cytotoxins’ modes of cytotoxicity and their impacts on inflammatory responses. In addition, we have submitted another accompanying review article, entitled “Pseudomonas aeruginosa: Taxonomy, Infections, Animal Modeling, and Conventional & Emerging Therapeutics” in which we provide a full section on current antibiotic treatments, and the mechanisms of P. aeruginosa resistance to them (including various efflux pumps). We have also included another section on emerging antibiotic-free approaches to control P. aeruginosa infections. The two review manuscripts were submitted to Cells at the same time.

  1. Discuss mechanisms that may be clinically applied to reduce the development of cytotoxicity and inflammatory responses in patients infected with P. aeruginosa.

Response: We appreciate the reviewer’s comments in this regard and have added a section in the revised manuscript to comply with the reviewer’s suggestion. Please see section 6 which is entitled “Cytotoxins as potential therapeutic targets.

Reviewer 2 Report

Dear authors,

The manuscript provides indeed an in-depth review of P. aeruginosa cytotoxic factors and their impact on host immune responses. The pathways of cytotoxin-induced programmed host cell death - e.g. apoptosis, pyroptosis and necroptosis, are depicted with impressive illustrations at molecular level. The cytotoxins of P. aeruginosa, their cytotoxicity modes and impacts on immune response are summarised in a detailed table. The presented information may be used for research and teaching purposes in both human and veterinary medicine, as this bacterium is of major importance among the opportunistic pathogens in both humans and animals.

Taking into account the significant number of cited references, some of them are not prepared strictly according to the journal format, e.g. journal names (abbreviated)  and volumes should be in italic. 

Author Response

We thank the reviewer for the thorough review of our manuscript and for his/her insightful comments. We are also very grateful to the reviewer for his/her recognition of the strengths of this review article. The reviewer indicated that some of our citations were not formatted according to the journal's formatting guidelines. We also noted that some journal names were abbreviated and some were not. We had used the Endnote software to insert and organize the citations. We had assumed that Endnote is complying with the journal’s formatting guidelines. With that said, we have corrected this issue in the revised manuscript.
